# An Intelligent Compaction Analyzer: A Versatile Platform for Real-Time Recording, Monitoring, and Analyzing of Road Material Compaction

**DOI:** 10.3390/s23177507

**Published:** 2023-08-29

**Authors:** Rajitha Ranasinghe, Arooran Sounthararajah, Jayantha Kodikara

**Affiliations:** SPARC Hub, Department of Civil Engineering, Monash University, Clayton Campus, Clayton, VIC 3800, Australia; rajitha.ranasinghe@monash.edu (R.R.); arooran.sounthararajah@monash.edu (A.S.)

**Keywords:** intelligent compaction, embedded system, accelerometer, intelligent compaction measurement values (ICMV), single-board computer (SBC), global navigation system (GNSS), real-time kinematic positioning (RTK), road compaction

## Abstract

Intelligent compaction (IC) is a technology that uses non-contact sensors to monitor and record the compaction level of geomaterials in real-time during road construction. However, current IC devices have several limitations: (i) they are unable to visualize or compare multiple intelligent compaction measurement values (ICMVs) in real-time during compaction; (ii) they are not retrofittable to different conventional rollers that exist in the field; (iii) they do not incorporate corrections for ICMVs reflecting variable field conditions; (iv) they are unable to integrate construction specifications as needed for performance-based compaction; and (v) they do not record all the key roller parameters for further compaction analysis. To address these issues, an innovative retrofittable platform with cutting-edge hardware and software was developed. This platform, called the intelligent compaction analyzer (ICA) platform, is effective at calculating conventional acceleration amplitude-based ICMVs and stiffness-based parameters and at displaying the spatial distributions of these parameters in a color-coded map in real-time during compaction.

## 1. Introduction

Intelligent compaction [1] (IC) is a process that utilizes vibratory rollers equipped with various sensors to automatically control and monitor the compaction level of geomaterials during road construction. The data gathered by these sensors is processed in real-time to display intelligent compaction measurement values (ICMVs), refs. [2,3,4,5,6,7,8,9], which are used to estimate the compaction state (i.e., density) of the material layer that is being compacted. The goal of IC is to improve the uniformity and consistency of the compaction process, leading to high-quality and long-lasting road pavements. In practice, newly laid road material layers are generally levelled using a grader and then compacted using a roller until the target density is reached. At the end of the compaction process, the density of the compacted layer is measured using spot density tests at randomly selected locations as part of the quality control and quality assurance process. Over the past few decades, various field studies have been undertaken to develop correlations between ICMVs and material density; however, an acceptable correlation has not yet been demonstrated for road materials [3,8].

Figure 1a depicts a schematic illustration of the current intelligent compaction (IC) technology. The controlling inputs of the compaction roller are the roller speed, roller travel direction, vibratory frequency, and vibratory amplitude. The response of the ground for a given set of system inputs is measured using the sensors attached to the roller. These sensors mainly include accelerometers [10,11] that measure the vibration acceleration, phase angle sensors that measure the position of the eccentric mass that excites vibration, and infrared (IR) temperature sensors [12,13] that measure the surface temperature of the compaction layer. The sensor inputs are collected by a system platform to generate an ICMV parameter. This value is then compared with the target value to calculate the difference, which is used by the roller operator to control roller compaction. In an ideal scenario, the system platform should be able to control the compaction process in the field by sending control signals to the roller. Figure 1b illustrates the main design steps involved in the development of the ICA platform. The main research problem highlighted in Figure 1 is that many existing compaction rollers in operation today lack the ability to measure in-built compaction levels, and those that exist do not have a unified compaction value or data acquisition methodology that results in diverse intelligent compaction measurement values (ICMVs) across different roller equipment manufacturers. Moreover, the same theoretical ICMV produces varying values across different device platforms when operating in parallel, largely due to discrepancies in noise filtering methods and data acquisition methodology. A notable void exists in the capability of current devices to record the raw data of the variables instrumental in computing compaction measurement values.

The design vision for the intelligent compaction analyzer device, as outlined in Figure 1, was conceived to bridge these gaps. This device is innovatively designed to retrofit onto conventional rollers, offering capabilities to measure varying compaction values, record essential raw data during the compaction process, and facilitate real-time processing along with visualization through a well-structured data processing pipeline. The design process began with defining system requirements, including selecting requisite sensors, computational power for real-time analysis, and essential communication protocols. Subsequent steps, all detailed in Figure 1, entailed the meticulous development of the software architecture, the choice of hardware components, the crafting of the sensor interface, and the formulation of reliable real-time processing algorithms. The user interface was designed for intuitive interaction, and communication capabilities were integrated, allowing the device to relay data to the cloud for deeper analysis. The design journey reached its apex with the transformation of development hardware into a robust, field-ready device, complete with a custom printed circuit board (PCB) and a sturdy enclosure. Rigorous testing and verification procedures ensured the refinement of the data processing pipeline and potential software feature adjustments.

Various manufacturers offer compaction measurement systems, such as the Trimble [14], TOPCON [15], Bomag [16], Caterpillar [17], Hamm [18], and Sakai [19]. These systems measure compaction level using non-contact sensors, display the results on built-in screens, and can also include GPS for location tracking and data storage [20]. However, many of these systems only come with new IC rollers and use different ICMVs, making it difficult to compare compaction results from different platforms. Trimble and TOPCON offer retrofittable devices; however, they only support basic ICMVs and do not provide flexibility to incorporate corrections for ICMVs.

The main objective of this study is to develop a versatile retrofit platform that can (i) capture multiple ICMVs and dynamic roller parameters in real-time during the compaction process, (ii) facilitate correction methods to reflect the variable field conditions, (iii) offer algorithm reconfiguration for evaluating the performance of different ICMVs, and (iv) monitor the raw vibration signal pattern and its frequency spectrum. The ICA platform has the following innovative features:Phase angle measurement of response vibration through the cross-correlation of dual axis acceleration signals: The compaction meters come with OEM rollers that utilize hall effect phase angle sensors that measure the movement of the eccentric mass mounted within the rotary drum of the compaction roller. The current retrofit devices lack the ability to calculate stiffness-based ICMVs as a phase angle sensor is non-retrofittable to a conventional roller. However, with the implementation of cross-correlation for dual-axis vibration measurement, the ICA platform is able to measure stiffness-based parameters (i.e., ${Sk}_{b}$ [2] and ${SE}_{vib}$ [21]) while remaining retrofit-friendly.A well-defined signal processing system: Through a process of trial and error, we identified the most suitable noise reduction algorithms and data sampling rates for the application of the ICA platform, such as the use of Hann Smoothing for noise reduction.The use of signal distortion measurement: We implemented the experimental algorithm called ‘MFD (modified fundamental distortion)’ to measure the distortion level of the excitation vibration force.The use of an autonomous vehicle grade GNSS system: The existing compaction measurement systems are equipped with survey-grade GNSS systems that were known to be slow in providing high-accuracy positional data, and they require bulkier GNSS antennas. However, the ICA platform leverages state-of-the-art GNSS technology [22] that can provide faster and more accurate positional data.Raw vibration signal visualization and data saving: While sophisticated compaction measurement calculation methods exist for extracting data from raw vibration signals, there are instances when it is beneficial to simply monitor the raw vibration data in real time and analyze it using fast Fourier transform (FFT). However, to the best of the authors’ knowledge, no existing compaction measurement systems offer this capability.Independent software analytic tools and cloud-based data storage and analysis: The ICA platform is equipped with its own compaction analysis page, and it has the capability to stream live data to the AWS (Amazon Web Services) cloud for storage and further analysis, which eliminates the need for a third-party software or physically copying data to a computer.

The paper is divided into four main sections. The Results section presents the outcomes of the ICA platform development and the results of the field experiments. The Development of the ICA platform section covers the development of the hardware, software, data processing pipeline, and data recording and streaming. The device functionality section provides an overview of the working flow of the platform. The Discussion section summarizes the key features of the ICA platform and its practical implications, and future works on the new technologies and concepts that will be implemented in the ICA platform.

## 2. Materials and Methods

### 2.1. Hardware Development

Figure 2a shows the hardware architecture of the ICA platform. The brain of the ICA platform is a single-board computer (SBC). An SBC consists of the central processing unit (CPU), random access memory (RAM), solid-state storage (SSS), and peripheral ports combined into a small form factor printed circuit board (PCB) [23,24,25,26,27]. The main two candidates for SBC were Nvidia Jetson Nano 4GB [28,29,30,31] and Raspberry Pi 4 Model B 8GB [32,33,34,35]. Nvidia Jetson Nano has a much more powerful graphics processing unit (GPU), which makes it ideal for computer vision and machine learning tasks, while the Raspberry Pi 4 Model B has the best overall computing power and a vast number of compatible modules and community support. The initial development was undertaken with Jetson Nano. However, the main data acquisition (DAQ) system was not compatible with the GPIO driver library of the Jetson Nano. Due to this driver incompatibility, Jetson Nano was ruled out and the Raspberry Pi 4 Model B 8 GB was selected as the SBC to implement the prototype ICA platform.

The data acquisition system for the acceleration data consists of dual MCC172 [36] boards stacked together where one is acting as backup. Each of the MCC172 boards can simultaneously sample two integrated electronics Piezo-electric (IEPE) [11] channels with a maximum sampling rate of 51.2 kHz with 24-bit resolution. A parallel sub-data acquisition system having an ADS1015 module [37] exists to capture analogue data. This system captures surface temperature data for the asphalt layer from a non-contact IR temperature sensor.

The geo-location system of ICA produces highly accurate GPS positions using ZED-F9R module [22]. It is capable of generating 0.2 m horizontal position accuracy (HPA) data with a navigation rate of up to 30 Hz when the real-time kinematic (RTK) [38,39,40] is available. NTRIP (networked transport of RTCM via internet protocol) data required for RTK is provided to the ZED-F9R module using the onboard Wi-Fi of SBC, as shown in Figure 2a.

**Remark** **1.**
*RTCM stands for Radio Technical Commission for Maritime Services. RTCM is an international non-profit organization that develops standards and protocols for various types of radio communication systems used in marine navigation, including those used for global navigation system (GNSS) applications. The most widely used RTCM standard is RTCM 3.x, which defines the format for transmitting differential corrections that can be used to improve the accuracy of GNSS positioning. Differential corrections are calculated by a reference station with known coordinates and transmitted to a user’s GNSS receiver, which uses the corrections to refine its own position estimate.*


Two global navigation system (GNSS) antennas [41] are available for the geo-location system, viz., TOP106 GNSS antenna [42] and magnetic GNSS antenna [43], and one of them can be selected on the basis of the GNSS signal strength at the site. The TOP106 GNSS antenna is a survey-grade antenna normally used when the GNSS signal strength is low. The magnetic GNSS antenna can be mounted on top of the metal roof of the roller cabin, which makes the installation process easy. This method was frequently used in our field trials. Specifications of each of these systems and the sensors are listed in Table 1. The costs for the hardware development are listed in Table 2.

The ICA platform can be operated using remote access and a local touch display. Parameters can be entered into the ICA using a wireless keyboard and a touchpad controller. Remote access to the ICA is provided through real virtual network computing (RealVNC) [44], and any mobile platform can be used to remotely monitor and control the ICA platform. The captured data are streamed to the amazon web services (AWS) cloud for backup. The data can be viewed remotely in real time using the web-based analytic dashboard powered by the AWS Grafana system. All the main hardware components, including the power supply system, backup battery, and electronics, were packed inside a 3D-printed enclosure, as shown in Figure 2b(ii).

### 2.2. Data Processing Pipeline

The ICA platform collects raw data from three data streams. The primary data stream is for collecting accelerometer data of two channels (vertical and horizontal), hereafter referred to as the IEPE DAQ system. The other two data streams collect the surface temperature and geo-location data. In this manuscript, we employ the term “data stream” to describe time-varying data, which are dynamically collected during the compaction process by the integrated sensors. These encompass the three data above streams. On the other hand, other constant data vital for calculations, including the roller parameters such as drum radius (Rd), drum width (*L*), static weight (Fc), the mass of the vibrating drum (md), and eccentric moment (mere), which are utilized to determine the stiffness-based values (Skb and SEvib) as mentioned in Section 2.2.1, are gathered through the user interface (UI) before the compaction process. These values are stored as constants in the program, facilitating real-time calculations. The entire data processing pipeline, including these additional details, is depicted in Figure 3. The data process cycle to generate ICMVs begins by collecting two raw voltage data profiles from the IEPE DAQ, respectively, proportional to the vertical and horizontal acceleration channel values.

Let *R* = Sample rate of the IEPE DAQ system.

Let *N* = Number of samples collected for processing at the given sampling rate.

The voltage data arrays collected for each vertical and horizontal acceleration channels in a process cycle are given as,
(1)$$\begin{array}{cc} V{\left(i\right)}_{ch0} & =\left\{v_{1},v_{2},v_{3},\ldots ,v_{i},\ldots ,v_{N}\right\}, \end{array}$$
where $i\in Z^{+}$ and ch0 is vertical component of acceleration.
(2)$$\begin{array}{cc} V{\left(i\right)}_{ch1} & =\left\{v_{1},v_{2},v_{3},\ldots ,v_{i},\ldots ,v_{N}\right\}, \end{array}$$
where $i\in Z^{+}$ and ch1 is horizontal component of acceleration.

As the next step, the vertical voltage data array is processed using a Hann window function [45,46,47] to perform Hann smoothing. The vertical component is chosen because it reflects the response of the compaction layer. Several considerations drive the choice of the Hann window function in this context:The Hann window tapers the data to zero at both ends, which is vital for accelerometer data. This tapering minimizes the abrupt transitions at the start and end of the signal segment, which can introduce unwanted high-frequency components [48].The Hann window finds an optimal balance when capturing frequency data. This balance ensures that the resulting frequency spectrum is neither smeared (losing resolution) nor riddled with spurious peaks (due to leakage) [49].When we compare it to other methods like Hamming or Blackman, the Hann window is a good choice. It works well for data with short, sudden spikes, like the data from accelerometers.

Applying the Hann smoothing technique to the voltage signal before the Fourier transform is pivotal. This step mitigates the spectral leakage that often manifests when executing a Fourier transform on imperfect periodic signals, ensuring the more precise identification of the actual frequency components.

As the next step, the vertical voltage data array is fed through a Hann window function [45,46,47] to perform Hann smoothing. The vertical component is selected as it is influenced by the response of the compaction layer. The Hann smoothing technique is applied to the voltage signal before performing Fourier transform to reduce the amount of spectral leakage that can occur when performing a Fourier transform on a non-perfect periodic signal.
(3)$$\begin{array}{cc} w(n) & =0.5\left(1-cos\left(2\pi \frac{n}{N}\right)\right),\;\mathrm{where}\;0\leq n\leq N \end{array}$$
(4)$$\begin{array}{cc} x(n) & =w\left(n\right)\times \left(\frac{V{\left(n\right)}_{ch0}}{V_{max}}\right),\;\mathrm{where}\;0\leq n\leq N,\;\mathrm{and}\;V_{max}=max\left(V(n)\right) \end{array}$$
Figure 3Data process pipeline.
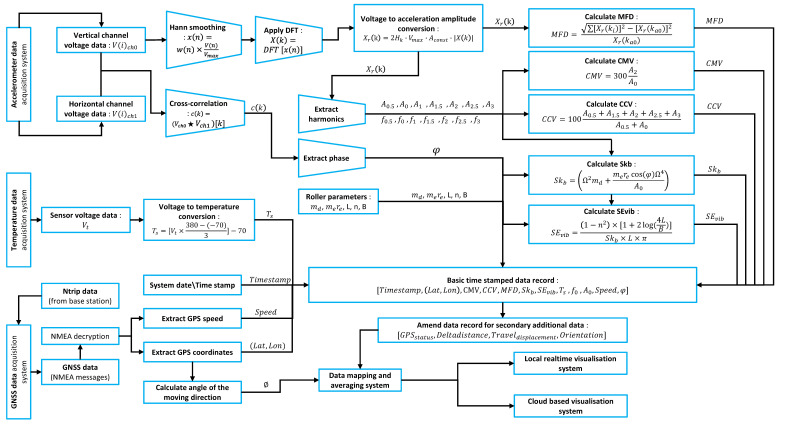


Then, one-dimensional discrete Fourier transform (DFT) is then performed on x(n) data array to obtain the frequency spectrum X(k).
(5)$$\begin{array}{cc} f & =\frac{k}{N}R \end{array}$$
(6)$$\begin{array}{cc} X(k) & =\sum_{n=0}^{N-1}x\left(n\right)\times e^{-i\frac{2\pi nk}{N}}, \end{array}$$
where *k* = index of the frequency bin and *f* is frequency.

Complex values in frequency spectrum X(k) are then converted to the real value spectrum $X_{r}\left(k\right)$ from the function below. To obtain the real amplitude values from a frequency spectrum obtained through the Fourier transform, absolute values are taken of the complex values ($\left|a+bi\right|=\sqrt{a^{2}+b^{2}}$) in the frequency spectrum. Furthermore, these values need to be normalized by the number of samples. Since the data come through a Hann filter, the resultant amplitude needs to be corrected by multiplying it with the Hann window compensation factor. Finally, the voltage amplitudes are converted to acceleration values by multiplying them with the accelerometer constant.
(7)Xr(k)=2×Hk×Vmax×Aconst×ReX(k)2+ImgX(k)2N,
where $H_{k}=2$, $A_{const}$ and $V_{max}$ are constants.

The parameter $A_{const}$ is constant for a given accelerometer sensor. $A_{const}$ converts voltage values to acceleration values. $H_{k}$ is the Hann window compensation factor. The fundamental frequency index is then determined from the maximum value of $X_{r}\left(k\right)$.
(8)$$\begin{array}{cc} A_{0} & =max\left(\left\{X_{r}\left(k\right):k=0,1,2\ldots max\left(k\right)\right\}\right), \end{array}$$
where $k_{a_{0}}$ is the frequency bin index of $A_{0}$.

The peak frequency bin index is quadratically interpolated to accurately calculate the fundamental frequency $f_{0}$.
(9)$$\begin{array}{cc} offset & =\sigma =\frac{2\left(2\times X_{r}\left(k_{a_{0}}\right)-X_{r}\left(k_{a_{0}}-1\right)-X_{r}\left(k_{a_{0}}+1\right)\right)}{X_{r}\left(k_{a_{0}}+1\right)-X_{r}\left(k_{a_{0}}-1\right)} \end{array}$$
(10)$$\begin{array}{cc} f_{0} & =\left(\left(k_{a_{0}}+\left(\sigma \right)\right)\times \frac{R}{N}\right) \end{array}$$

The harmonic frequencies and amplitudes are then calculated using the fundamental frequency $f_{0}$.
(11)$$\begin{array}{cc} k_{i} & =\frac{i\times f_{o}\times N}{R}+0.5, \end{array}$$
where $i\in \left\{0.5,1,1.5,2,2.5,3\right\}$, $k_{i}$ is rounded up for the nearest positive integer $k_{i}\in Z^{+}$.
(12)$$\begin{array}{cc} A_{i} & =X_{r}\left(k_{i}\right) \end{array}$$
(13)$$\begin{array}{cc} f_{i} & =\frac{k_{i}}{N}R \end{array}$$
where $f_{i}$ and $A_{i}$ are the harmonic frequencies and acceleration amplitudes, respectively. In this system, the sample rate of the IEPE DAQ is R=51.2 kHz, and the number of samples recorded is N=7000. The parameter *N* is a user input variable from the platform UI. After processing the raw values into meaningful data, the next step is calculating different ICMV parameters.

#### 2.2.1. Calculation of Compaction Measurement Parameters

In a data process cycle, the first value to be calculated from the processed raw data is the modified fundamental distortion (MFD) for acceleration.
(14)$$\begin{array}{cc} MFD & =\frac{\sqrt{\sum_{i=0}^{}{\left(X_{r}\left(k_{i}\right)\right)}^{2}-{\left(X_{r}\left(k_{a_{0}}\right)\right)}^{2}}}{X_{r}\left(k_{a_{0}}\right)}, \end{array}$$
where $i\in Z^{+}$ and $k_{i}\in Z^{+}$.

The MFD is calculated as the ratio between the RMS values of all discrete frequency brackets over the fundamental amplitude in the ICA platform. This is similar to the well-known total harmonic distortion (THD) value in electrical engineering; however, for MFD calculation, all the available frequency brackets are considered, whereas THD considers only the harmonics of the fundamental frequency. MFD directly represents how much the fundamental frequency (primary excitation vibration of the drum roller) is distorted by other vibrations and harmonics.

For calculating the stiffness parameters of the compaction material layer, the phase angle between the excitation force of the roller drum and the response from the compaction material layer is required. Since the ICA is a retrofit platform, an indirect method is used to estimate this phase angle. The phase angle between the horizontal $V{\left(i\right)}_{ch0}$ and vertical $V{\left(i\right)}_{ch1}$ channel components is calculated by applying the cross-correlation to these two signals [50,51,52,53]. This involves normalizing the two signals by subtracting the mean and dividing by the standard deviation.

Let $\bar{V{\left(i\right)}_{ch0}}$, $\sigma_{ch0}$ be the mean value and standard deviation of $V{\left(i\right)}_{ch0}$.

Let $\bar{V{\left(i\right)}_{ch1}}$, $\sigma_{ch1}$ be the mean value and standard deviation of $V{\left(i\right)}_{ch1}$.
(15)$$\begin{array}{cc} V{\left(i\right)}_{ch0}^{n} & =\frac{V{\left(i\right)}_{ch0}-\bar{V{\left(i\right)}_{ch0}}}{\sigma_{ch0}},\;\mathrm{where}\;i\in Z^{+},\;n=\;\mathrm{normalized}\;\mathrm{signal} \end{array}$$
(16)$$\begin{array}{cc} V{\left(i\right)}_{ch1}^{n} & =\frac{V{\left(i\right)}_{ch1}-\bar{V{\left(i\right)}_{ch0}}}{\sigma_{ch1}},\;\mathrm{where}\;i\in Z^{+},\;n=\;\mathrm{normalized}\;\mathrm{signal} \end{array}$$

Here, $V{\left(i\right)}_{ch0}^{n}$ and $V{\left(i\right)}_{ch1}^{n}$ are normalized signals of vertical and horizontal acceleration channels. These dual normalized channels are used for cross-correlation c(k) calculation. The phase angle is then calculated from the time delay taken for the argmax of the cross-correlation.
(17)$$\begin{array}{cc} c(k) & =\sum_{n}^{}V{\left(n+k\right)}_{ch0}^{n}.\bar{V{\left(n\right)}_{ch1}^{n}} \end{array}$$
(18)$$\begin{array}{cc} dt & =\frac{1}{R} \end{array}$$
(19)$$\begin{array}{cc} t_{max} & =N\times dt \end{array}$$
(20)$$\begin{array}{cc} t{\left(i\right)}_{array} & =(\left\{-t_{max},\ldots ,0,\ldots ,t_{max}\}\right),\;\mathrm{where}\;0\leq i\leq 2N-1\;\mathrm{and}\;i\in Z^{+} \end{array}$$
(21)$$\begin{array}{cc} Phase & =\varphi =\frac{t{\left(argmax\left(c(k)\right)\right)}_{array}}{dt}\times 360 \end{array}$$

The final part of a data process cycle is to calculate acceleration and phase-dependent compaction measurement parameters. The ICA platform calculates parameters CMV, CCV, $Sk_{b}$, and $SE_{vib}$ using the following equations.
(22)$$\begin{array}{cc} CMV & =300\times \frac{A_{2}}{A_{0}} \end{array}$$
(23)$$\begin{array}{cc} CCV & =100\times \frac{A_{0.5}+A_{1.5}+A_{2}+A_{2.5}+A_{3}}{A_{0.5}+A_{0}}, \end{array}$$
where $A_{i}$ values are harmonic amplitudes $i\in \left\{0.5,1,1.5,2,2.5,3\right\}$.
(24)$$\begin{array}{cc} \Omega^{2} & ={\left(2\times \pi \times f_{0}\right)}^{2} \end{array}$$
(25)$$\begin{array}{cc} Sk_{b} & =\left(\Omega^{2}m_{d}+\frac{m_{e}r_{e}\cos\left(\varphi \right)\Omega^{4}}{A_{0}}\right)\times 10^{-6}, \end{array}$$
where $m_{d}$ is the mass of the vibrating drum and $m_{e}r_{e}$ is the eccentric moment.
(26)$$\begin{array}{cc} B & =\sqrt{\frac{16\times R_{d}\left(1-n^{2}\right)\times F_{c}}{\pi \times SE_{vib}\times L}}, \end{array}$$
where $R_{d}$ is the drum radius, *L* is the drum width, and $F_{c}$—is the static weight.
(27)$$\begin{array}{cc} SE_{vib} & =\frac{\left(1-n^{2}\right)\times \left(1+2\log(\frac{4L}{B})\right)}{Sk_{b}\times L\times \pi }, \end{array}$$
where *n* is the compaction material Poisson’s ratio.

Equations of $SE_{vib}$ and *B* are solved iteratively within each data acquisition process cycle.

#### 2.2.2. Geolocation of Data

The raw data collected from the GNSS system of the ICA platform includes GPS location (latitude and longitude) (Lat,Lon) and the speed of the roller $U_{km/h}$. A 2D array is generated to map the measured parameters (ICMVs, temperature, and passcount) based on their latitude and longitude coordinates, where each index in the array corresponds to a specific pixel on the map. This allows us to easily visualize and analyze data in relation to specific geographical locations. These 2D arrays or 2D pixel maps are unique for each compaction area, and their height and width constants are determined by the boundary coordinates of the area. The relationship between GPS locations and pixels is then calculated to map the measured GPS locations to the 2D pixel map. A user-defined constant value is used to convert distances into a number of pixels (the pixel resolution per meter).

Let $M_{i}\in R^{H\times W}$ be a 2D pixel map, where *H* is height and *W* is width constants unique for each compaction area.

To determine the *H* and *W* of the 2D pixel map, the boundary coordinates of the compaction area are required. The relationship between pixels and GPS locations is calculated as follows.

Let $\left(lat_{j},lon_{j}\right)$ be the GPS location measured in real-time using the GNSS system of the ICA platform.

Let $B_{i}=\left\{\left(lat_{1},lon_{1}\right),\ldots \ldots ,\left(lat_{K},lon_{K}\right)\right\}$ be the boundary coordinate array, where *K* is the number of boundary points obtained to cover the compaction area.

The constraints of the 2D pixel maps are then defined as follows:(28)$$\begin{array}{cc} Lat_{max} & =max(B_{i}) \end{array}$$(29)$$\begin{array}{cc} Lat_{min} & =min(B_{i}) \end{array}$$(30)$$\begin{array}{cc} Lon_{max} & =max(B_{i}) \end{array}$$(31)$$\begin{array}{cc} Lon_{min} & =min(B_{i}) \end{array}$$(32)$$\begin{array}{cc} \varphi & =cos(\frac{Lat_{max}+Lat_{min}}{2}) \end{array}$$

The measured GPS locations are then converted into distances concerning the minimum boundary latitude and longitude.
(33)$$\begin{array}{cc} dx & =(lat_{j}-Lat_{min})\times 111,120 \end{array}$$
(34)$$\begin{array}{cc} dy & =(lon_{j}-Lon_{min})\times \varphi \times 111,120,\;\mathrm{where}\;dx\;\mathrm{and}\;dy\;\mathrm{are}\;\mathrm{lengths}\;\mathrm{in}\;\mathrm{meters} \end{array}$$

A user-defined constant PPM (pixels per meter) value is used to convert distances into a number of pixels. The position of the measured GPS location $\left(lat_{j},lon_{j}\right)$ is then mapped to the 2D pixel map $M_{i}$ with reference to its origin coordinate $\left(x_{0},y_{0}\right)$.
(35)$$\begin{array}{cc} pdx & =dx\times PPM \end{array}$$
(36)$$\begin{array}{cc} pdy & =dy\times PPM,\;\mathrm{where}\;pdx\in Z^{+}\;\mathrm{and}\;pdy\in Z^{+} \end{array}$$
(37)$$\begin{array}{cc} \left(x,y\right) & =(x_{0}+pdx,y_{0}+pdy),\;\mathrm{where}\;\left(x,y\right)\;\mathrm{are}\;\mathrm{pixel}\;\mathrm{coordinates} \end{array}$$

The width of the 2D pixel map $M_{i}$ is defined as W=max(x), and the height is defined as H=max(y). The reverse of the same equations is used to calculate GPS coordinates back from the pixel coordinates.

The 2D pixel maps provide three functions in the ICA platform. First, they are used to generate a color-coded map to visualize the variation of compaction parameters (an example is shown in Figure 2b(iii,iv) and Figure 4). Second, each cell in the 2D pixel map averages its values in real-time. Third, the 2D pixel maps are used to analyze data in the field without any post-processing although the ICA platform internally works in the 2D domain. The collected data are also recorded as time-stamped data records and are saved in the **.csv* file format for post analyzing. Such recorded data can easily be analyzed to display the quality and uniformity of material compaction. A collection of field test data records are shown in Figure 4 and Figure 5.

### 2.3. Software Development

The software of ICA platform was developed to be lightweight and embedded platform friendly [54,55]. The UI was developed using PySide2, which has the lesser general public license (LGPL) python module of Qt framework [56]. All other programming library dependencies are shown in Table A1 in the appendix. ICA software consists of 20 classes, of which 9 are UI classes, as shown in Figure 6. The main class inherits from the QMainWindow class and has instances of all other UI classes. Project class, running class, analyze class, DAQ class, GPSsetup class, WifiSetup class, roller class, environment class, and material class inherit from QWidget class of PySide2. These UI classes share information with each other by calling the class instances through the main class. Figure 2b(iii,iv) shows the visualization of main user interfaces.

#### Software Functionality

All classes of ICA can be categorized into four sections (global methods and algorithms, project initialization and analyzing, collect static parameters, and collect and process dynamic parameters) according to their purpose as in Figure 6. Methods, calculation, and ICData classes provide equations and common methods. Instances of these classes are accessible through the main class. Material, environment and roller classes provide interfaces to gather parameters needed for ICMV calculations. These classes are also responsible for saving and loading these parameters to **.csv* files for future use.

WifiSetup, DAQSetup, and GPSsetup classes are responsible for raw data acquisition, pre-processing, and connectivity. DAQSetup provides UIs to monitor raw data and the frequency spectrum of the collected data. It also collects sensor settings related to accelerometers ($A_{const}$, noise threshold, input channel, and number of samples). The back end of the DAQSetup UI is handled by the daq_data class. This class handles communication with the hardware responsible for acquiring temperature and acceleration data. The DAQSetup class is also responsible for recording raw acceleration and sensor settings and saving them as **.csv* files if the user commands it.

The GPSsetup class is responsible for providing accurate GPS location data using RTK technology. The UI collects NTRIP broadcaster connection settings from the user. The backend of gps_data class uses user-defined NTRIP broadcaster settings to establish a GNSS data stream connection. It then continuously collects RTCM data from the data stream and feeds it to the GNSS module to correct the GPS data. This data connection is executed on a separate CPU process to avoid a time lag in the main process cycle. NtripClient and serialCOM classes provide helper methods for the gps_data class to establish this GNSS data stream.

WifiSetup UI collects the user’s WiFi SSID (service set identifier) and password and connects to the local WiFi. Internet connection through WiFi is a requirement of the ICA platform because GPS RTK and remote monitoring depend on it. However, the ICA platform can be used without internet connectivity, but this will lead to low GPS accuracy, and a local interface is required for platform control. The WifiSetup UI class also uses a back-end Wifi class that handles the communication of commands to the operating system.

The project, running, and analyze UI classes are the most active user interfaces. These interfaces collect initial conditions, run the data collection process, visualize the data in real-time, and perform post-analysis. In each run, the ICA software starts from the project page. Project UI collects preliminary data required for new data collection. A grid parameter, which is the inverse of PPM, is taken as user input. Also, the boundary coordinate array $B_{i}$ is taken from the project page and can be saved for later use.

The running class handles the Running page, which is shown in Figure 2b(iii). The running page facilitates the run-time visualization of data, saving data and setting manual direction, lane number, and section flags for the data record. The purpose of these flags is to enter information manually by the user to ease the post analyzing. The running interface provides selection buttons for each ICMV parameter, and the selected data map is shown with the respective user-defined color map. The running class also collects all the data from other class instances in run-time and combines them to for a single data record. This data record is then saved to a **.csv* file, as mentioned in the data processing pipeline section. Each data collection process starts when the ‘set map’ button is pressed. This generates empty 2D pixel maps, $M_{i}$, using the boundary coordinates $B_{i}$ and starts parallel processes for the GNSS system and the DAQ system, respectively. When the ‘start’ button is pressed, the calculation and data populating the 2D pixel map will be initiated. When the ‘save’ button is pressed, the collected data will be saved in the user-defined project folder without stopping the data collection process. The running class uses the back-end GPSMAP class to visualize 2D pixel maps. This class inherits components from PyQtGraph library GraphicsLayoutWidget.

The analyze class handles the data analysis user interface (UI), which is used to browse the data records with respect to the 2D pixel map. This interface also provides an ROI (region of interest) tool to select a region to zoom and provide average data values for that region. A real-world example is shown in Figure 2b(iv). The visualization of the analyze class also inherits components from the PyQtGraph library.

A demonstration video of the ICA platform and its software is available in the Appendix A.

### 2.4. Data Recording and Streaming

The ICA platform has a well-structured local data-saving hierarchy, as shown in Figure 7. All the parameters collected from the user interfaces are saved separately in **.csv* format in the *settings* folder according to their modality. Boundary coordinates of the compaction area are saved in the main folder, which shares the project name. *csvData* folder contain two **.csv* files. The file named *RawAccelerationData* contains raw accelerometer values. This file is saved only if the user requests it by ticking the *Raw* check box in the Daq data settings interface. The file named *rawdata* contains recorded and processed data with time stamps, and the file has 21 data columns that contain raw and processed data shown in Table 3. Each data record is streamed to a cloud database in real-time for backup and further analytics.

The *data* folder contains the **.pkl* format file, which holds all the 2D pixel maps. The *datalist* file and the *config* file in the main folder are generic data files that contain internal parameters (PPM, $A_{const}$, noise threshold, input channel, etc.) related to the project.

## 3. Results

### 3.1. The ICA Platform

The latest version of the ICA platform is shown in Figure 2. The ICA platform is a complete system that integrates data capturing, processing, analyzing, and displaying. The key hardware components of the ICA platform are shown in Figure 2a. These are explained in detail in the hardware development section. Figure 2b(i) shows how the field tests are carried out. The data extracted from the field tests are shown in Figure 4 and Figure 5, which are explained in the field demonstration results section. Figure 2b(ii) shows the prototype ICA platform used for field tests. The ICA platform has both a local display and a web-based analytic dashboard. The local display helps to guide the roller operator in real-time during compaction, and the web-based tool (Figure 2b(v)) can be used to monitor the compaction process remotely. The sensors are connected to the ICA device with BNC connectors [57], and the platform provides a high-definition multimedia interface (HDMI) port for an external monitor and several universal serial bus (USB) ports for transferring data and peripheral attachment. Figure 2b(iii,iv) show the ICA’s main local user interfaces (UIs). The roller parameters and settings required for compaction data processing are entered using these local device user interfaces, and they also provide real-time data monitoring and analyzing capabilities. The local display is visible to the roller operator during the compaction process. The details of other user interfaces are provided in the Appendix A. The ICA software saves both raw and processed data for further analysis purposes in the local device memory and streams the data into the AWS cloud in real time for remote monitoring and backup.

### 3.2. Field Demonstration Results

Figure 5 shows an example of field data captured by the ICA platform during the pre-mapping of an existing asphalt pavement. Pre-mapping is defined as rolling the compaction roller at a low vibration setting on the existing road pavement before placing a new material layer. A zoomed-in compilation of all recorded data for *lane A* (lane names are marked in Figure 5a), which includes roller dynamic data (i.e., speed, travel displacement, fundamental frequency, and fundamental amplitude), ICMV values (i.e., CMV [58], CCV [59], ${Sk}_{b}$ [2], and ${SE}_{vib}$ [21]), the surface temperature, and MFD for acceleration are shown in Figure 4. The data recorded by the ICA platform were immaculate, well within the theoretical limits and responsive to varying ground conditions. The recorded data demonstrate the effectiveness of the ICA platform when comparing multiple ICMV parameters. The ICMV maps produced in color-coded format for the compaction area (Figure 4) could be used to analyze the following parameters:Non-uniform compaction areas: The ICMV maps can identify non-uniform compaction areas in the material layer, which may be caused by variations in the material properties, layer thickness, or roller passes. These areas may require additional compaction to achieve the desired compaction quality. In Figure 4a,b, the CMV and CCV values are irregular and the CMV and CCV maps show non-uniformity. However, in Figure 4c,d the ${SE}_{vib}$ and MFD values are consistent and uniform for the same compaction area, which highlights the existence of a poor correlation between current ICMVs used in practice.Comparison with historical data: The ICMV maps can also be used to compare the current compaction results with historical data to track the progress of the compaction over time. This can help the roller operator to make adjustments to the compaction parameters to achieve the desired compaction quality as per the design specifications.Correlation with compaction quality: The ICMV maps are useful to evaluate the quality of the compaction of the material layer. By comparing the ICMV values with the design specifications, the operator can identify the areas that require additional compaction to achieve the desired density or stiffness.Verification of roller coverage (or roller pass count map): The ICMV maps can also be used to verify the coverage of the roller passes over the compacted area. These maps should show a consistent distribution of roller passes across the entire compaction area. The areas with low ICMVs may indicate insufficient compaction.

Suthakaran et al. [8] assessed the efficiency of this ICA platform in identifying areas of inadequate compaction by utilizing real-time ICMV data generated during the compaction of an asphalt overlay testbed. The ICA platform was instrumented on a dual-drum vibratory roller for this study. The findings revealed a remarkable correlation between variations in asphalt density, as measured by a nuclear density gauge (NDG) at ten distinct locations within the testbed, and the ICMV data (both before compaction—premapping and during compaction), along with the asphalt temperature data provided by the ICA platform.

## 4. Discussion

The main aim of this study was to develop an innovative retrofittable platform (named the ‘ICA’ platform) with cutting-edge hardware and software tools to address the key limitations of the current IC technology. The ICA platform can capture 21 dynamic parameters, including 5 ICMVs that have not been recorded in one platform in the past. In addition, the platform is (i) in modular form, making it retrofittable to an existing conventional roller; (ii) configurable to incorporate correction methods for different ICMV parameters reflecting variable field conditions; and (iii) customizable to road construction specifications towards performance-based IC while digitalizing the parameters that govern the quality and uniformity of the compaction for the entire construction corridor.

In this paper, we have provided the basic design concepts of the ICA platform, its functionalities, and its capabilities, with initial field validation results. As shown in the results, the processed data from the field experiments were sufficiently clean and reflected the variations in the compaction material layer. The ICA platform was able to calculate conventional acceleration amplitude-based ICMVs (i.e., CMV and CCV) and stiffness-based parameters (i.e., $SE_{vib}$, ${Sk}_{b}$) and display the spatial distributions of these parameters in a color-coded map in real-time during compaction. It should be noted that the analysis of various ICMVs with respect to the density of the compaction material layer was not presented in this paper. The hardware, software, and analytics need to be further refined in the future based on the field trials undertaken on different road materials using different roller types.

### Future Works

In the real-time compaction monitoring process, the use of inclination sensors to measure the compaction surface level has not yet been explored in the ICA platform. We plan to integrate inclination sensing into the ICA device in the future and explore how the inclination data can be used together with vibration-based compaction measurement parameters to achieve performance-based construction. We have filed a patent on the innovation of a technique that involves the measurement of deformation of the compaction layers during roller passes to retrieve density evolution using a laser sensor array (International Patent Application No. PCT/AU2021/051505) [60]. The relevant sensing hardware and analytics will be integrated into the ICA platform. The sensing hardware will be refined and versatile so that the validated technology can be used with an existing conventional roller by retrofitting [61,62]. Further, data analytics will be developed to produce spatial moisture maps of the compaction area using a ground penetrating radar (GPR). The estimated moisture data will be transferred to the ICA platform in real time through a cloud-based system. During the compaction process, the ICA platform will display the spatial distributions of density and moisture content in a color-coded map along with ICMVs. Utilizing these data, advanced analytics will be developed and implemented into the ICA platform to achieve performance-based compaction control needed for intelligent pavement construction in line with the industrial 4.0 revolution.

## Figures and Tables

**Figure 1 sensors-23-07507-f001:**
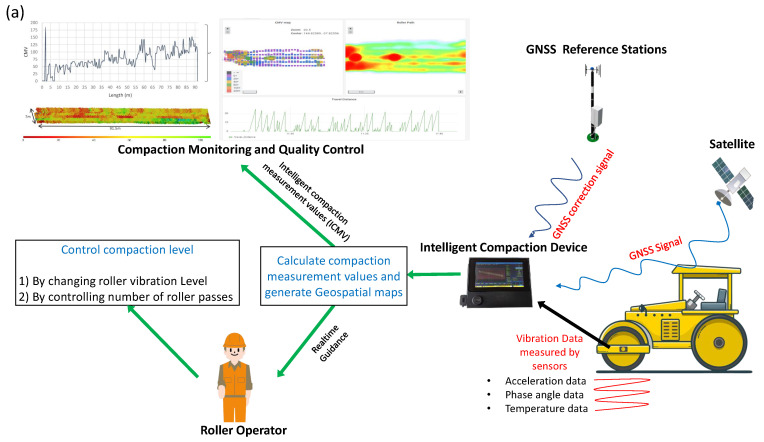
(**a**) Illustration of intelligent compaction (IC) technology. The ICMVs are parameters used to measure the compaction level of road material layers during roller compaction in the field. During the compaction process, the compaction roller is controlled by an operator to achieve the targeted degree of compaction for the road material. The roller operator obtains real-time visualization of the compaction from the compaction measurement system. (**b**) The design of the intelligent compaction analyzer (ICA) device aims to overcome several drawbacks of existing compaction measurement systems as listed in the illustration. The design process starts with targeting specific problems. Hardware and software are developed through a process of trial and error to overcome the drawbacks in field conditions.

**Figure 2 sensors-23-07507-f002:**
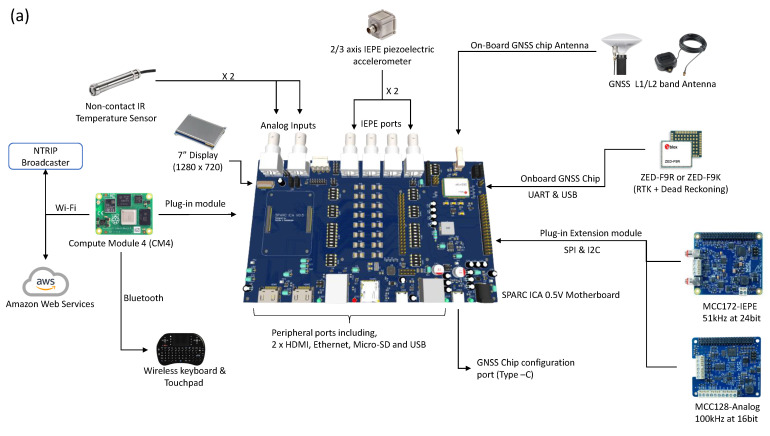
(**a**) Hardware architecture of the ICA platform. (**b**-**i**) Field testing of the ICA platform for initial data validation. (**b**-**ii**) Prototype of the ICA platform (Version 0.4). The IEPE accelerometer ports connect accelerometers, which capture vertical and horizontal accelerations. The IR probe port connects a non-contact IR temperature sensor. (**b**-**iii**) Mapping user interface, which controls the data acquisition process that provides a real-time map of the compaction material layer with instantaneous values of roller speed, frequency, amplitude, and surface temperature of the asphalt layer for a selected ICMV parameter. (**b**-**iv**) Data analyzing user interface that is used to view data records along with the final compaction map. The details of the other user interfaces are provided in the Appendix A. (**b**-**v**) Web-based analytic interface powered by Amazon web services (AWS) Grafana.

**Figure 4 sensors-23-07507-f004:**
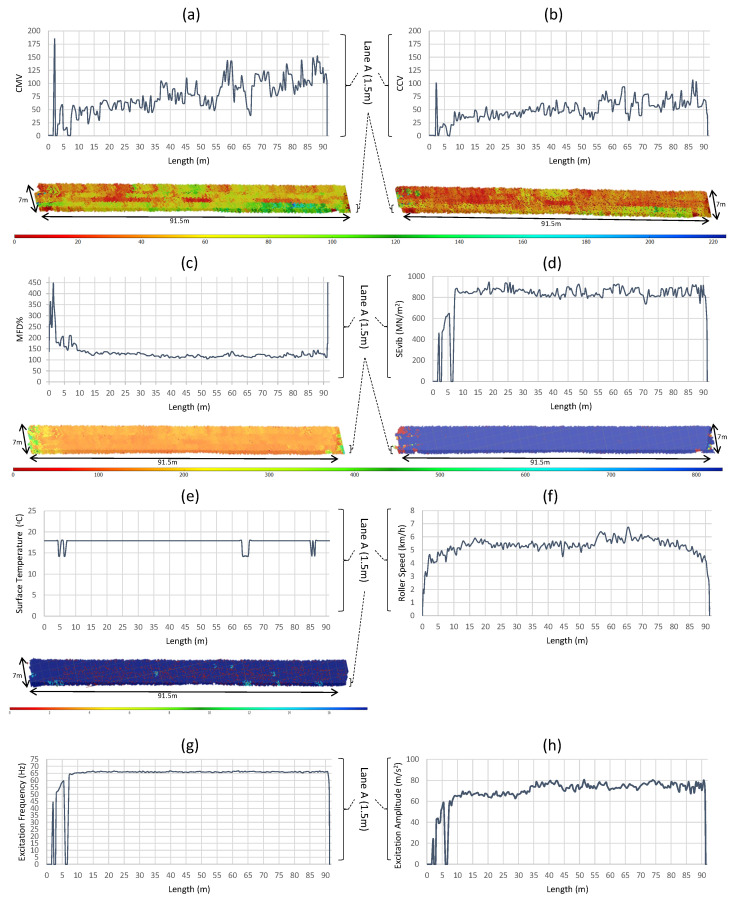
Each plot corresponds to a 1.5 m lane (lane A as shown in Figure 5) in a 7 m width asphalt pavement. (**a**–**e**) show CMV, CCV, MFD, and ${SE}_{vib}$ (Equations (14), (22), (23), and (27), respectively) ICMV and surface temperature plots with their color-coded maps for lane A, respectively. (**f**–**h**) show roller speed, frequency, and amplitude for lane A, respectively. The performance of the ICA platform was examined during the compaction of soil and unbound granular materials, and similar illustrations are provided for these materials in the Appendix A.

**Figure 5 sensors-23-07507-f005:**
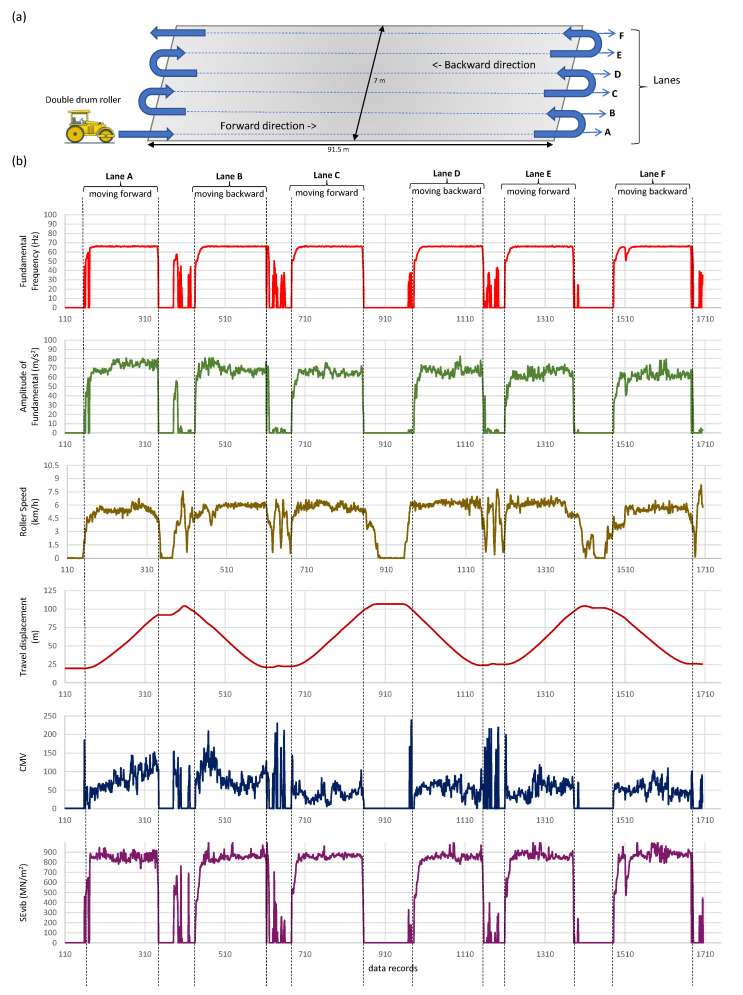
Illustration (**a**) schematically shows the roller compaction pattern in an actual road construction project. Each lane (A, B, C, D, E, and F) has about 33% overlap. Plots in (**b**) show the data envelope for the main parameters concerning the roller compaction pattern in (**a**). CMV and ${SE}_{vib}$ are defined in Equations (22) and (27), respectively.

**Figure 6 sensors-23-07507-f006:**
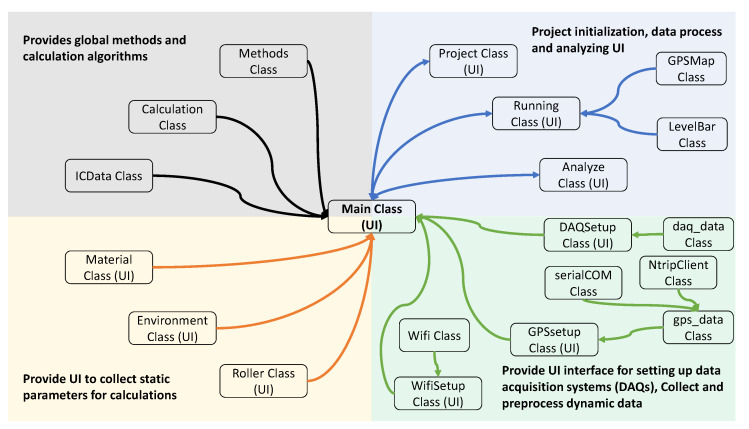
High-level software class architecture diagram.

**Figure 7 sensors-23-07507-f007:**
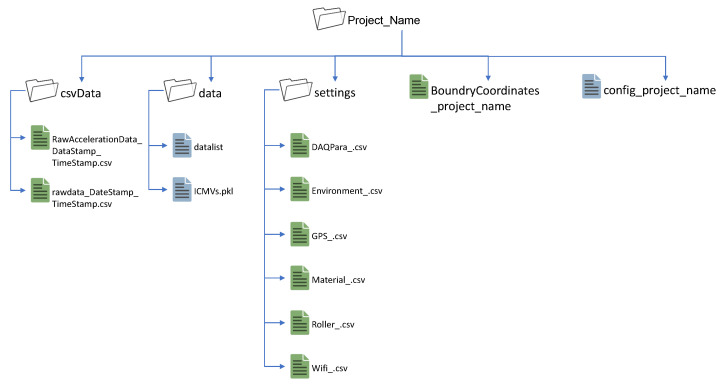
File saving hierarchy.

**Table 1 sensors-23-07507-t001:** Hardware specification and sensor resolutions.

Purpose	System Specification	Sensor Specification
Measure vertical and horizontal vibration profile of the main vibratory drum.	Dual channel simultaneous sampling rate of 51.2 kHz with 24-bit resolution.	Measuring range: ±140 ms${}^{-2}$, Type: Triaxial IEPE accelerometer, Sensitivity: <5% uncertainty error below 1 kHz.
Measure surface temperature when compacting asphalt.	Max sampling rate of 3.3 kHz at 12-bit resolution	Measuring range: −70 °C, Type: IR temperature sensor, Accuracy: ±0.5 °C∼±4 °C, FOV: 5${}^{\circ }$
High-precision geo-position measurement. Vehicle speed data measurement.	Horizontal position accuracy of 0.20 m with GPS-RTK technology. Max navigation data rate up to 30 Hz. Velocity accuracy of 0.5 m/s. Heading accuracy of 0.2 degrees.	ZED-F9R high precision dead reckoning module. L1, L2/L5 frequency band magnetic GNSS antenna.

**Table 2 sensors-23-07507-t002:** Hardware cost breakdown of ICA platform.

System	Component	Cost (AUD)
Motherboard of ICA device	Custom made printed circuit board (PCB)	116.96
	Assembling and component cost	350.85
GNSS system	ZED-F9R module	422.43
	GNSS antenna	139.95
Vibration data acquisition system	2 × MCC172 (IEPE data acquisition module) module	2389.44
	2 × HBK triaxial DeltaTron IEPE accelerometer, brackets, and cables	6394.00
Temperature data acquisition system	MCC128 (analog data acquisition module)	466.84
	TS01 non-contact IR temperature sensor	144.95
Enclosure	3D prints	240.00
Miscellaneous hardware	Battery, power supply system, cables, ports, and peripherals	1200.00
Total hardware cost		11,865.42

**Table 3 sensors-23-07507-t003:** Data records.

Data Column Name (as of **.csv* File)	Unit or Format	Description
Time_Stamp	MM/DD/YYYY HH:MM:SS	Time stamp for each data record
GPS_Latitude	Floating point number	Latitude in decimal degrees
GPS_Longitude	Floating point number	Longitude in decimal degrees
GPS_Status	Fixed RTK, Float RTK, DGNSS or 2D/3D	GPS accuracy level
CMV	Floating point number	Compaction meter value
CCV	Floating point number	Compaction control value
MFD	Floating point number	Modified fundamental distortion
$Sk_{b}$	MN/m	Stiffness parameter
$SE_{vib}$	MN/m${}^{2}$	Vibratory modulus
Surface_Temperature	Celsius	Surface temperature
FundamentalFreq	Hz	Frequency of the vibratory roller
FundamentalAmplitude	ms${}^{-2}$	Amplitude of the vibratory roller
Speed	km/h	Speed of the vibratory roller
Delta_Distance	m	Distance traveled between two data points
Direction	S, F or B	Relative travel direction: S = Stop, F = Forward, and B = Backward
Map_Origin	[GPS_Latitude, GPS_Longitude]	GPS coordinate where data recording started
Travel_displacement	m	Displacement from the Map_Origin
Phase_Estimate	Degrees	Estimated phase angle
Orientation	F or B	Manual input for roller direction
Lane	A,B, etc.	Manual input for compaction lane
Section	0,1,2 etc.	Manual input for compaction area/section

## Data Availability

The raw datasets used and/or analyzed during the current study are available from the corresponding author upon reasonable request. The processed data are included in this published article and its Appendix A.

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
