# Peer review of "An Intelligent Compaction Analyzer: A Versatile Platform for Real-Time Recording, Monitoring, and Analyzing of Road Material Compaction"

_sensors, 2023, doi:10.3390/s23177507_

Round 1

Reviewer 1 Report

The paper proposed a novel system for compaction monitoring and analysis of road materials, which overcomes various shortcomings in existing research, including the inability to monitor multiple compaction indicators in real-time, limited adaptability to different field conditions, issues with the correction of monitoring values, and the inability to record roller parameters. The authors presented an integrated system and device that can record and analyze parameters based on acceleration and stiffness, while considering roller parameters, demonstrating good innovation. It is of great interest and significance. Therefore, it can be recommmended for publication consideration. However, there are still some minor issues that need to be addressed:

1. In the concluding paragraph of the introduction section (lines 89-97), the alignment between the cited references and the subsequent subheadings is inconsistent. It is important to ensure clearness and consistency in both order and content between the cited references and the subheadings in order to facilitate the readers' understanding of the main structure of the paper.

2. In the section on data processing flow (2.2), it mentioned that there are three main data streams: the first data stream primarily consists of acceleration-related data, while the second and third data streams are surface temperature and positioning data, respectively. However, it is unclear how the data processing flow includes the stiffness-related data and the roller-related parameters. Please clarify how these data were incorporated into the processing flow or provide an explanation how such data were handled.

3. In the field test example section (3.2), there was a lack of comparison with reliable validation results or alternative methods. Are there any measures taken to validate the effectiveness of the proposed integrated device in data processing for construction purposes? It would be beneficial to include relevant validation measures to ensure that the newly proposed integrated device meets the requirements of construction to some extent.

Quality of English Language is acceptable. The writing is very fluent and professional; however, some minor grammatical checks and typo-corrections may still need.

Reviewer 2 Report

Very interesting subject.

Figure 1 b) is referred at the end of item 1. This figure should be better presented and explained.

Reviewer 3 Report

Please fix the formatting issues for the tables and equations. Everything else looks good.

Reviewer 4 Report

This manuscript tends to introduce a device design of an ICA platform for intelligent compaction. The topic is of interest. However, there are some comments below for consideration:

1) In general, this manuscript reads more like a technical report for a device design. It includes a lot of details on device design, however, not very organized, and a little hard to follow.

2) Introduction needs to be rewritten with sufficient literature review. The current introduction, especially figure 1 is too general and the words in figure 1 is too small, cannot be seen very well. It is suggested to have a thorough literature review instead of using a figure to describe the literature review.

3) There are quite some functions or parameters were selected without explaining why those parameters were selected that way, for instance, in line 159, a hann window function, why?

4) Table 2 and Table 3 have some format issue.

5) This manuscript focuses on introducing the design of the device, but lack explaining why and how. 

6) The results section is very weak and lack of explanations and comparation with other current available methods. Please include more on the data collected and how to use to show the developed device is better?
